# Isomerization of Asp7 in Beta-Amyloid Enhances Inhibition of the α7 Nicotinic Receptor and Promotes Neurotoxicity

**DOI:** 10.3390/cells8080771

**Published:** 2019-07-25

**Authors:** Evgeny P. Barykin, Alexandra I. Garifulina, Elena V. Kruykova, Ekaterina N. Spirova, Anastasia A. Anashkina, Alexei A. Adzhubei, Irina V. Shelukhina, Igor E. Kasheverov, Vladimir A. Mitkevich, Sergey A. Kozin, Michael Hollmann, Victor I. Tsetlin, Alexander A. Makarov

**Affiliations:** 1Engelhardt Institute of Molecular Biology, Russian Academy of Sciences, Vavilov St. 32, 119991 Moscow, Russia; 2Shemyakin-Ovchinnikov Institute of Bioorganic Chemistry, Russian Academy of Sciences, Miklukho-Maklaya Street, 16/10, 117997 Moscow, Russia; 3Sechenov First Moscow State Medical University, Institute of Molecular Medicine, Trubetskaya Street 8, bld. 2, 119991 Moscow, Russia; 4Department of Biochemistry I – Receptor Biochemistry, Ruhr University, 44780 Bochum, Germany

**Keywords:** amyloid-beta, nicotinic acetylcholine receptor, modifications, Alzheimer’s disease, neurotoxicity, calcium imaging, radioligand analysis, aspartate isomerization

## Abstract

Cholinergic dysfunction in Alzheimer’s disease (AD) can be mediated by the neuronal α7 nicotinic acetylcholine receptor (α7nAChR). Beta-amyloid peptide (Aβ) binds to the α7nAChR, disrupting the receptor’s function and causing neurotoxicity. In vivo not only Aβ but also its modified forms can drive AD pathogenesis. One of these forms, iso-Aβ (containing an isomerized Asp7 residue), shows an increased neurotoxicity in vitro and stimulates amyloidogenesis *in vivo*. We suggested that such effects of iso-Aβ are α7nAChR-dependent. Here, using calcium imaging and electrophysiology, we found that iso-Aβ is a more potent inhibitor of the α7nAChR-mediated calcium current than unmodified Aβ. However, Asp7 isomerization eliminated the ability of Aβ to decrease the α7nAChR levels. These data indicate differences in the interaction of the peptides with the α7nAChR, which we demonstrated using computer modeling. Neither Aβ nor iso-Aβ competed with ^125^I-α-bungarotoxin for binding to the orthosteric site of the receptor, suggesting the allosteric binging mode of the peptides. Further we found that increased neurotoxicity of iso-Aβ was mediated by the α7nAChR. Thus, the isomerization of Asp7 enhances the inhibitory effect of Aβ on the functional activity of the α7nAChR, which may be an important factor in the disruption of the cholinergic system in AD.

## 1. Introduction

The accumulation of data on acetylcholine deficiency and the decrease of acetylcholinesterase (AChE) [1,2] in the brain of patients with Alzheimer’s disease (AD) provided the basis for the cholinergic hypothesis of AD [3]. The decrease in acetylcholine levels and AChE activity in the cortex and hippocampus observed in patients with AD may be caused by the selective death of cholinergic neurons [4]. AChE inhibitors as stimulants of the cholinergic system are one of the few drugs that alleviate AD symptoms, confirming the importance of cholinergic insufficiency in shaping the clinical picture of the disease [5]. The degeneration of cholinergic neurons may be caused by the interaction of beta-amyloid peptide (Aβ) with nicotinic acetylcholine receptors (nAChRs), which are predominantly expressed in neurons of this type [6,7]. The two most abundant types of nAChRs in the human brain are heteromeric α4β2 and homomeric α7, both involved in the regulation of sleep, pain, appetite, and a number of cognitive functions [7,8,9,10]. Recent data indicate that α7nAChR is a promising target for AD therapy [11]. The α7nAChR binds to Aβ with picomolar affinity, which leads to the internalization of the Aβ–α7nAChR complex [12] and can induce neurodegeneration [13] and plaque formation [14]. It is known that plaque formation and neurodegeneration can be triggered by structurally and chemically modified forms of Aβ [15,16,17,18]. One of the most common chemical modifications of Aβ is the isomerization of Asp7 residue (iso-Aβ) [19,20,21]. Isomerization of Asp7 is present in about 50% of Aβ molecules in amyloid plaque cores [19,22]. Recent data confirm that isomerized Aβ accumulates with aging and is elevated in AD [23]. There is a clear association of iso-Aβ with amyloidogenesis: injections of iso-Aβ aggravate amyloidosis in the brain of model mice, and an increased accumulation of iso-Aβ in brain tissues of AD patients has been shown [23,24]. It can be assumed that the interaction of the α7nAChR with iso-Aβ plays an important role in the disruption of cholinergic transmission and death of cholinergic neurons. In this paper, using bioinformatic approaches, calcium imaging, electrophysiology, and radioligand analysis, we studied the effects of isomerization of Asp7in Aβ on its interaction with the α7nAChR. We have shown that the isomerization of Asp7 enhances the inhibitory effect of Aβ on the functional activity of the α7nAChR, which may be the reason for the increased neurotoxicity of the peptide and an important factor in the pathogenesis of AD.

## 2. Materials and Methods

### 2.1. Preparation of Aβ Peptides

#### Synthetic Peptides: Aβ_42_


[H2N]-DAEFRHDSGYEVHHQKLVFFAEDVGSNKGAIIGLMVGGVVIA-[COOH] and iso-Aβ_42_, were obtained from Biopeptide (San Diego, USA). Peptides were monomerized as described previously [25]. A fresh 5 mM solution of Aβ was prepared by adding 10 µl of 100% anhydrous dimethyl sulfoxide (DMSO) (“Sigma”) to 0.224 mg of peptide, followed by incubation for 1 h at room temperature to completely dissolve the peptide.

### 2.2. Mouse Neuroblastoma Cell Culture and Transient Transfection

Mouse neuroblastoma Neuro2a cells obtained from the Russian collection of cell cultures (Institute of Cytology, Russian Academy of Sciences, Saint Petersburg, Russia)were cultured in DMEM Dulbecco Modified Eagle Medium (DMEM) (Paneco, Moscow, Russia) supplemented with 10% FBS. They were sub-cultured the day before transfection and were plated at a density of 10,000 cells per well in a 96-well black plate. On the next day Neuro2a cells were transiently transfected with plasmids coding human α7nAChR (α7nAChR-pCEP4), a fluorescent calcium sensor Case12 (pCase12-cyto vector, Evrogen, Moscow, Russia) and chaperone NACHO (TMEM35-pCMV6-XL5, OriGene, USA) in a molar ratio of 4:1 following a lipofectamine transfection protocol (ThermoFisher Scientific, Waltham, MA, USA). Transfection with NACHO significantly increases the expression level of this receptor [26]. The transfected cells were grown at 37 °C in a CO_2_ incubator for 48–72 h, before performing the calcium imaging assay. Cells transfected with Case12 and NACHO, but not with α7nAChR were used to calculate background.

### 2.3. Radioligand Assay

Mouse neuroblastoma Neuro2a cells transiently transfected with human α7nAChR were pre-incubated for 72 h with DMSO (final concentration 0.2%) or Aβ_42_ or iso-Aβ_42_ (10 μM dissolved in DMSO) in culture medium. DMSO and the peptides were applied to cells in serum-free DMEM, and bovine serum to 10% was added 2 h later. After that, 5 × 10^5^ cells were incubated with 1.5 nM radioiodinated α-bungarotoxin (^125^I-αBgt) (500 Ci/mmol) in 50 μL of DMEM (Paneco, Russia) supplemented with protease inhibitors (PMSF, Sigma). After incubation for 30 min at room temperature, radioligand binding was stopped by filtering the incubation mixture through Whatman GF/F filters presoaked in 0.5% polyethylenimine. Then the filters were washed 3× with 4 mL cold 20 mM Tris-HCl, 0.01% BSA (pH 8.0), and bound radioactivity was measured in a Wallac 1470 Wizard Gamma Counter (PerkinElmer, Waltham, MA, USA). To determine non-specific ^125^I-αBgt binding, a 200x excess of α-cobratoxin [27] was added to the control samples. Specific binding was calculated as the difference between total and nonspecific binding.

### 2.4. Competition Radioligand Assay

For a competition binding assay we used (1) the heterologously expressed acetylcholine binding protein (AChBP) from *Lymnaea stagnalis* or *Aplysia californica* or (2) the α7 nAChR-transfected human cell line GH4C1. AChBP from *A. californica* and *L. stagnalis* was kindly provided by Prof. S. Luo, Key Laboratory for Marine Drugs of Haikou, Hainan University, China. The rat pituitary tumor-derived cell line GH4C1, which stably expresses human α7 nAChR was received from Eli Lilly and Company, London, UK, and used immediately after defrosting without cultivation or passaging. ^125^I-αBgt, which binds to AChBP and α7 nAChR with 150 and 0.5 nM affinities, respectively, was used as the radioligand. AChBP and GH4C1 cellswere incubated in 50 μL of binding buffer (20 mM Tris-HCl buffer, pH 8.0, containing 1 mg/mL BSA) for 40 min with 15 μM of amyloid peptides, followed by a 5-min incubation with 1 nM ^125^I-αBgt (500 Ci/mmol). Prior to the radioactivity measurements, GH4C1 samples were filtered through glass GF/C filters (Whatman) pretreated with 0.3% polyethylenimine and washed thrice with cold 20 mM Tris-HCl buffer, pH 8.0, containing 0.1 mg/mL BSA. The AChBPs samples were filtered through DE-81 filters presoaked in phosphate-buffered saline containing 0.7 mg/mL BSA and washed thrice with the same buffer. Bound radioactivity was measured as described above.

### 2.5. Human Neuroblastoma SH-SY5Y Culture and Differentiation

Undifferentiated human neuroblastoma cells SH-SY5Y, obtained from ATCC, were cultured in DMEM/F12 medium (ThermoFisher Scientific, Waltham, MA, USA) supplemented with 10% fetal bovine serum (FBS) (PAA Laboratories GmbH, Pasching, Austria), 2.5 µg/mL amphotericin B and 50 µg/mL gentamicin in a CO_2_ incubator at 37 °C and 5% CO_2_ atmosphere. For differentiation, cells were sub-cultured and plated at a density of 5000–10,000 cells per well in a 96-well black plate (Corning Inc., Corning, NY, USA). *All-trans* retinoic acid (RA) (Sigma) was added the day after plating at a final concentration of 10 µM in DMEM/F12 with 10% bovine serum. After 5 days of incubation with RA, cells were washed once with serum-free DMEM/F12 and incubated with 50 ng/mL brain-derived neurotrophic factor (Sigma) in serum-free DMEM/F12 for 3 days [28].

### 2.6. Calcium Imaging

Calcium imaging in N2a and SH-SY5Y cells was performed according to previously published protocols [29,30]. To detect the human α7nAChR-mediated intracellular calcium rise, Case12-transfected Neuro2a cells were incubated with its positive allosteric modulator (PAM) PNU120596 [31] (10 µM, Tocris Bioscience, Bristol, UK) for 20 min at room temperature before agonist addition. SH-SY5Y cells expressing human α7nAChRs natively were loaded with a fluorescent dye Fluo-4, AM (1.824 μM, ThermoFisher Scientific, Waltham, MA, USA) and a water-soluble probenecid (1.25 mM, ThermoFisher Scientific, Waltham, MA, USA) for 30 min at 37 °C and then were kept for 30 min at room temperature according to the manufacturer’s protocol. Then the SH-SY5Y cells were accordingly incubated with PNU120596 for 20 min before agonist addition. Transfected Neuro2a cells expressing α7nAChRs and SH-SY5Y cells expressing α7nAChRs natively were pre-incubated with Aβ_42_ or iso-Aβ_42_ (10 µM) for 30 min at room temperature before agonist addition. α7nAChR-dependent intracellular calcium rise was induced with the highly specific α7nAChR agonist PNU282987 [32]. Pre-incubation with 15 µM of α-cobratoxin, a specific inhibitor of the α7nAChR function [27] (purified from *Naja kaouthia* venom), for 15 min was used as a control. Measurements were performed as described previously [29,30]. Data files were analyzed using Hidex Sense software (Hidex, Turku, Finland).

### 2.7. Electrophysiology

Rat α7nAChR in the pSGEM vector were linearized using XbaI (Promega, USA). mRNAs were transcribed in vitro using the T7 mMessage mMachine (Ambion Inc., Austin, TX, USA) transcription kit. RNAs were purified by phenol:chloroform extraction and isopropanol precipitation. Stage V±VI *Xenopus laevis* oocytes were defolliculated with 2 mg/mL collagenase Type I (Life Technologies, USA) at room temperature (21–24 °C) for 2 h in Barth’s solution without calcium (88.0 mM NaCl, 1.1 mM KCl, 2.4 mM NaHCO_3_, 0.8 mM MgSO_4_, 15.0 mM HEPES/NaOH, pH 7.6) for 1.5 ± 2 h at 20 °C. The oocytes were stored in Barth’s solution with calcium (88.0 mM NaCl, 1.1 mM KCl, 2.4 mM NaHCO_3_, 0.3 mM Ca(NO_3_)_2_, 0.4 mM CaCl2, 0.8 mM MgSO_4_, 15.0 mM HEPES/NaOH, pH 7.6) supplemented with 63.0 μg/mL penicillin-G sodium salt, 40.0 μg/mL streptomycin sulfate, 40.0 μg/mL gentamicin.

Oocytes were selected and injected with 5 ng cRNA of rat α7nAChR in a total injection volume of 15 nL. After the injection, the oocytes were incubated at 18 °C in ND96 buffer or in Barth’s solution with calcium for 48 ± 120 h. Electrophysiological recordings were made using a Turbo TEC-03X amplifier (Npi electronic, Germany) and Patch master software (HEKA, Germany), at a holding potential of −60 mV. Oocytes were placed in a small recording chamber with a working volume of 50 μL, and 50–100 μL of agonist (acetylcholine) solution in ND96 electrophysiological buffer or Ba^2+^ Ringer’s solution (115.0 mM NaCl, 2.5 mM KCl, 1.8 mM BaCl_2_, 10.0 mM HEPES/NaOH, pH 7.2) were applied to an oocyte. Oocytes expressing rat α7nAChR were pre-incubated with Aβ_42_ or iso-Aβ_42_ (10 µM) for 3 min followed by its co-application with acetylcholine (3–1000 µM). To allow receptor recovery from desensitization, the oocytes were superfused for 5 ± 10 min with buffer (1 mL/min) between the ligand applications.

### 2.8. Modelling of Aβ:α7nAChR and Iso-Aβ:α7nAChR Interaction

Modelling of the Aβ_42_ three-dimensional structure has been performed using templates selected from a survey of Aβ structures in the PDB database which we carried out earlier [33]. We have also utilized results of ab initio modeling with the Bhageerath server [34]. The subsequent expert modeling and energy minimization was followed by computation of molecular dynamics (MD) trajectories for the obtained Aβ_42_ structure, yielding the final model structure [35]. 

The three-dimensional structure of the iso-Aβ_42_ has been modeled applying a similar MD technique. Prior to this, modelling of the isomerized D7 residue in the Aβ_42_structure has been performed. The α7nAChR extracellular domain structure has been modeled by the Swiss Model server using the structures of α7nAChR-AChBP chimeras 5AFJ and 3SQ6 from the Protein Data Bank as the templates.

Modeling of interaction was carried out by global docking using the QASDOM meta-server (http://qasdom.eimb.ru/) developed by us [36]. Sets of Aβ_42_ and iso-Aβ_42_ models interacting with the extracellular domain of α7nAChRwere obtained by global docking using Gramm-x, ClusPro, SwarmDock, and Zdockweb-servers.

### 2.9. Neurotoxicity Measurements

Human neuroblastoma cells SH-SY5Y were differentiated as described above in 96-well plates. Differentiated cells were incubated with 10 µM of Aβ_42_ or iso-Aβ_42_ for 72 h in serum-free media in the presence of 20 ng/mL BDNF. After the incubation, cells were washed with PBS and stained with either 1 µM EthD-1 (Thermo Scientific) for 30 min at 37 °C or WST reagent (Sigma), diluted in culture media 1:10 for 2 h at 37 °C. EthD fluorescence and absorbance of WST was measured with a Spark microplate reader (Tecan).

### 2.10. Statistical Methods Used for Data Analysis

Data are presented as means of at least three independent experiments ± SD or SE. The comparison of data groups in toxicity tests and in radioligand assay was performed using one-way ANOVA with post-hoc testing (using unpaired samples Student’s *t*-test with Bonferroni correction); after a Bonferroni correction a *p* value < 0.016 was considered as statistically significant. Statistical analysis was performed using STATISTICA 8.0 (StatSoft Inc., Tulsa, OK, USA) and OriginPro 9.0 software (OriginLab, Northampton, MA, USA).

## 3. Results

### 3.1. Effects of Aβ_42_ and Iso-Aβ_42_ on Functional Activity of α7nAChR in N2a and SH-SY5Y Cells and in Xenopus Laevis Oocytes

To study the effects of Aβ_42_ and iso-Aβ_42_ on the α7nAChR, we used two cell lines expressing α7nAChR. The first one, the human neuroblastoma cell line SH-SY5Y, expresses α7nAChR endogenously. To achieve neuron-like expression patterns and morphology, it was differentiated as described above. The second line, mouse neuroblastoma N2a, was co-transfected with α7nAChR and the chaperone NACHO. The efficiency of the transfection in N2a cells was confirmed with fluorescent α-bungarotoxin staining (Appendix A) as described previously [30]. Receptor function studies were performed with the calcium imaging technique. We observed a robust response of both cell lines to α7nAChR stimulation and the determined half maximal effective concentration (EC50) for the agonist PNU282987 was close to published values (Appendix A) [36]. The kinetics and the amplitude of the response did not depend on the calcium sensor used (fluo-4 in SH-SY5Y or Case12 in N2a) (Appendix A), which correlates with the previously published data, where we have shown the equivalence of fluo-4 and Case12 in N2a for analysis of both the α7nAChR and the muscle-type nAChR functional activity [30].

Pre-incubation with Aβ_42_ or iso-Aβ_42_ for 30 min significantly suppressed the intracellular calcium rise in SH-SY5Y and α7nAChR-transfected N2a cells, induced by PNU282987 (Figure 1A,B). We did not observe any response in α7nAChR-untransfected N2a cells, and the application of α-cobratoxin completely suppressed the calcium rise in both SH-SY5Y and α7nAChR-transfected N2a cells, indicating that the observed response was mediated by the α7nAChR [27]. The effects of Aβ_42_ and iso-Aβ_42_ were different in both the intensity of the inhibition and in their inhibition pattern. In N2a cells, the Aβ_42_ peptide inhibited the receptor response by 30% at the maximum concentration of PNU282987, whereas for iso-Aβ_42_ the degree of inhibition reached 60%. In addition, the inhibitory effect of the peptides differed sharply depending on the concentration of PNU282987. Thus, inhibition of the α7nAChR by the Aβ_42_ peptide was not observed at 30 or 60 nM of PNU282987, whereas iso-Aβ_42_ inhibited the receptor response by 90% at these concentrations of PNU282987, as compared with the control (Figure 1A). In the SH-SY5Y cells, the inhibition of the α7nAChR by the Aβ_42_ and iso-Aβ_42_ peptides did not differ at the maximum concentration of PNU282987; however, at low concentrations of the agonist, we observed differences in the action of the peptides similar to those obtained in the N2a cells (Figure 1B). In both cell lines, the incubation with iso-Aβ_42_ increased the EC50 of the α7nAChR for PNU282987, while Aβ_42_ had virtually no effect (in N2a cells) or even decreased (in SH-SY5Y cells) the EC50 (Table 1).

The effects of Aβ_42_ and iso-Aβ_42_ were also analyzed by two-electrode voltage clamp in *X. laevis* frog oocytes expressing the α7nAChR. The application of acetylcholine to oocytes pre-incubated with the amyloid peptides for 3 min showed an inhibition of the currents by both Aβ_42_ and iso-Aβ_42_ (Figure 1C). The effects of the peptides observed in the oocytes were almost identical, with the exception of the slightly larger (by 10%) inhibitory effect of iso-Aβ_42_ at the maximum acetylcholine concentration used (1 mM). The inhibitory effect of the amyloid peptides on the α7nAChR in *X. laevis* oocytes, as in the N2a cells, increased along with the increasing concentration of the agonist. The observed responses in the oocytes were characteristic for the α7nAChR, and no ACh-induced currents were detected in the α7nAChR cRNA-uninjected oocytes (Figure 1D).

### 3.2. Aβ_42_ and Iso-Aβ_42_ Do not Compete with α-Bungarotoxin for Binding to the α7nAChR or to Acetylcholine-Binding Proteins (AChBPs)

The differences in the affinity of the amyloid peptides for the α7nAChRwere studied with a competitive radioligand assay using radioiodinated α-bungarotoxin (^125^I-αBgt). The ability of Aβ_42_ and iso-Aβ_42_ at a concentration of 15 μM to inhibit ^125^I-αBgt binding was studied on the α7nAChR expressed in GH4C1 cells and on purified recombinant acetylcholine-binding proteins (AChBPs) from *Aplysia californica* and *Lymnaea stagnalis*. Neither Aβ_42_ nor iso-Aβ_42_ inhibited the binding of ^125^I-αBgt to the AChBPs (Appendix A) after 40 min of pre-incubation. In addition, no significant inhibition by the amyloid peptides of ^125^I-αBgt binding to the α7nAChR expressed by the GH4C1 cells was detected (Appendix A).

### 3.3. The Aβ_42_-Induced Reduction of α7nAChR Representation is Neutralized by the Isomerization of Asp7

Since we observed no inhibition of the binding of ^125^I-αBgt to the α7nAChR or AChBPs after 40 min of pre-incubation with the amyloid peptides, we decided to test how a prolonged incubation with Aβ_42_ or iso-Aβ_42_ would affect the level of the α7nAChR. It has previously been established that prolonged incubation with Aβ_42_ results in a decrease in the α7nAChR expression in PC12 neuroblastoma cells [37]. To test the effect of the peptides on the receptor levels, we used N2a neuroblastoma cells, which transiently express the α7nAChR. The incubation of the N2a cells with Aβ_42_ for 72 h resulted in a 35% decrease in the specific binding of ^125^I-αBgt to the α7nAChR (Figure 2), indicating a decrease in the α7nAChR levels in N2a cells. The incubation with iso-Aβ_42_ did not affect the amount of ^125^I-αBgt bound to the cells.

### 3.4. Comparison of the Toxicity of Aβ_42_and Iso-Aβ_42_ toward SH-SY5Y Cells

The effects of Aβ_42_ and iso-Aβ_42_ on the α7nAChR differed not only for the short-term but also for the long-term incubation of cells with the peptides. It is known that the α7nAChR can mediate the toxic effect of Aβ_42_ on neuronal cells [38], so we tested how the isomerization of Asp7 changes the α7nAChR-mediated cytotoxicity of Aβ_42_. To this end, we examined the neurotoxic effects of Aβ_42_ and iso-Aβ_42_on differentiated SH-SY5Y neuroblastoma cells in the presence and absence of the selective α7nAChR inhibitor α-bungarotoxin (αBgt). The incubation of SH-SY5Y cells with Aβ_42_ for 72 h resulted in a 25% increase in necrotic cells (Figure 3A) and a 5–7% decrease in cell viability (Figure 3B). The iso-Aβ_42_ showed a higher neurotoxicity with a 60% increase in the number of necrotic cells, and the cell viability decreased by 25%. The toxic effect of αBgt at a concentration of 50 nM was similar in magnitude to the effect of iso-Aβ_42_. The co-application of Aβ_42_ with αBgt caused a toxic effect similar to that of iso-Aβ_42_ (Figure 3B, in the middle), while the simultaneous addition of iso-Aβ_42_ and αBgt did not change the toxicity (Figure 3A,B, righthand columns). The higher neurotoxicity of iso-Aβ_42_ compared to that of Aβ_42_ was not due to its enhanced aggregation or differences in oligomeric composition (Appendix A), as we demonstrated by the electrophoretic analysis of covalently linked low-molecular weight oligomers obtained by photo-induced cross-linking (Supplementary Methods) [39].

### 3.5. Molecular Modeling of the Binding Sites for Iso-Aβ_42_ and Aβ_42_ in the α7nAChR

To obtain additional data on how Asp7 isomerization affects the interaction of Aβ_42_ with the α7nAChR, we constructed the models of peptide–receptor complexes using molecular modeling. Using the Aβ_42_ structure as the initial model [33], the iso-Aβ_42_ model was created. The modeling showed that the isomerization of Asp7 significantly changed the structure of the peptide (Figure 4A). The sets of Aβ_42_ and iso-Aβ_42_ complexes with the α7nAChR extracellular domain obtained on Gramm-x, ClusPro, SwarmDock, and Zdock servers were analyzed using the QASDOM meta-server. The modeling of the receptor–peptide complexes indicated the presence of common binding sites for Aβ_42_ and iso-Aβ_42_ (inside the receptor channel), as well as different binding sites on the outer surface of the receptor (Figure 4B). An analysis of the probability of the interaction of the peptides with different α7nAChR amino acid residues (Appendix A) showed that the predicted peptide binding sites overlap with the acetylcholine binding site and the known allosteric binding sites [40,41]. For both Aβ_42_ and iso-Aβ_42_, bindings to Loop E and Loop F of the agonist-binding pocket were predicted. By contrast, a strong binding in the Loop C region was observed for Aβ_42_ but was completely absent for iso-Aβ_42_. For both peptides we predicted binding in the regions of 6–15 and 64–72, the amino acid residues of which participate in the formation of the “top pocket” allosteric site, and in the region of 97–103, which forms part of the “vestibule pocket” allosteric site [41], with the binding of Aβ_42_ in the “top pocket” being much stronger (Appendix A). On the other hand, binding to the L38 residue, which is part of another allosteric site, the “agonist subpocket”, was predicted for iso-Aβ_42_ only [41]. For both peptides, an interaction with the Trp55 residue, which is critical for the rapid desensitization of the receptor [42], was predicted. In the resulting models, both Aβ_42_ and iso-Aβ_42_ demonstrated an interaction in the region of residues 55–61. It has previously been shown that a peptide corresponding to these residues is able to destroy the Aβ complex with the receptor [43]. Another peptide with similar properties corresponds to residues 146–155, and in this area we predicted binding for Aβ_42_ but not for iso-Aβ_42_. In areas not related to known binding sites, with the exception of the area of 129–142, interaction was not observed. In the area of 129–142, which corresponds to the distal (upper) part of the receptor (Figure 4B), binding was predicted only for iso-Aβ_42_.

Amino acid residue-wise analysis of Aβ_42_ and iso-Aβ_42_ interaction with the receptor suggested a possible basis for the differences in the inhibitory effects of the peptides (Appendix A). Compared to Aβ_42_, binding of iso-Aβ_42_ to the α7nAChR is significantly reduced for the residues 7–10 of the peptide, probably reflecting a sharp turn (Figure 4A) occurring in the peptide secondary structure due to the Asp7 isomerization. The hydrophobic C-terminus of iso-Aβ_42_, comprising residues 27 till 40, forms an inward loop, which leads to weaker binding of this region to the α7nAChR than in Aβ_42_. On the contrary, in iso-Aβ_42_ stronger binding to the α7nAChR was predicted around the residues 11–16 and for the residue Phe20.

## 4. Discussion

The disruption of cholinergic transmission in AD was one of the first detected phenomena that determine the clinical picture of the disease [44]. Loss of nicotinic receptors in the brain tissues of patients has been observed with both a reduction in the number of ligand binding sites and a reduction of the levels of the receptor subunits [45]. The α7nAChR is well-presented in brain tissue and is an important component of the cholinergic system involved in memory formation [14]. It is known that the Aβ_42_ peptide is able to bind to the α7nAChR with high affinity [13], which leads to the inhibition of the receptor [46], neuron death [13], and may precede the formation of plaques [14].

In the sporadic form of AD, neurodegeneration and amyloidogenesis can be induced by modified forms of Aβ [15,16,17,18]. One of these forms is iso-Aβ, which has a stronger neurotoxic and amyloidogenic effect than Aβ [24,47].

We compared the inhibitory properties of iso-Aβ_42_ and Aβ_42_ toward the α7nAChR. For the measurements, we used the α7nAChR transiently expressed in N2a cells [30] and in *X. laevis* oocytes [48], or endogenously expressed in differentiated SH-SY5Y cells [49,50]. Previously, we utilized the α7nAChR-transfected N2a cells and α7nAChR cRNA-injected *X. laevis* oocytes to analyze the interaction of the receptor with various natural [51] and synthetic [52] low-molecular compounds, neurotoxic peptides and proteins [53], as well as with human toxin-like three-finger proteins [48,54]. Both Aβ_42_ and iso-Aβ_42_ inhibited intracellular calcium rise caused by α7nAChR activation in differentiated SH-SY5Y cells or in N2a cells (Figure 1). However, iso-Aβ_42_ showed a greater suppression of the receptor-induced increase in the concentration of intracellular Ca^2+^ (Figure 1A) and, unlike Aβ_42_, the effect of iso-Aβ_42_ did not disappear with a decrease in the concentration of the agonist (Figure 1A,B). Thus, the isomerization of Asp7 leads to an increase in the intensity of inhibition and a change in the nature of the inhibitory effect of Aβ_42_ on the α7nAChR. To confirm the interaction between the α7nAChR and the peptides, we measured the effects of iso-Aβ_42_ and Aβ_42_ in α7nAChR-transfected *X. laevis* oocytes. The oocytes represent a pure system without factors affecting the interaction, such as the presence of the other nicotinic receptors or the positive allosteric modulator PNU120596. The use of PNU120596 in calcium imaging is inevitable since it amplifies agonist-induced α7nAChRs responses to the detectable level. PNU120596 increases the probability of transient α7nAChR activation by agonists, and also destabilizes a ligand-bound non-conducting “desensitized” state of the receptor [55,56,57]. In oocytes, both iso-Aβ_42_ and Aβ_42_ exhibited an inhibitory effect on the α7nAChR, however, these effects hardly differed (Figure 1C). Such similarity can be attributed to the absence of secondary effects in oocytes due to α7nAChR internalization, alteration of membrane potential, or induced by Ca^2+^ entry. The electrophysiology was performed in the absence of Ca^2+^ in Ba^2+^-containing buffer, which also prevents current inactivation and receptor desensitization due to acute Ca^2+^ rise [58,59,60]. In contrast to previously published data [13,61], our results show that both Aβ_42_ and iso-Aβ_42_ are noncompetitive α7nAChR inhibitors that bind at the allosteric site. Both in experiments on cell lines and in *X. laevis* oocytes, incubation with peptides reduced the maximum response of the receptor, which became even more pronounced with increasing concentrations of the agonist, indicating a non-competitive inhibition. In addition, we observed an increase in the inhibitory effect of peptides with an increase in the concentration of the agonist, which is also a property of allosteric inhibitors [48,62]. The allosteric nature of the binding was confirmed by the lack of competition between αBgt, the orthosteric inhibitor of α7nAChR, and the amyloid peptides both in binding to an AChBP and to the α7nAChR in GH4C1 cells (Appendix A).

Despite the lack of competition between the amyloid peptides and αBgt during a 40 min incubation, the incubation of N2a cells with Aβ_42_ for 72 h caused a 30% decrease in αBgt binding to the cells (Figure 2). Since Aβ does not compete with αBgt, this effect was due to a 30% decrease in α7nAChR levels in the N2a cells. According to the literature, the formation of the Aβ–α7nAChR complex leads to the endocytosis of the peptide–receptor complex [63] and a decrease in the level of the α7nAChR [37]. Taking into account the stronger inhibitory effect of iso-Aβ_42_ on the receptor, we expected to see the same or more pronounced effect of this peptide on the expression of the α7nAChR. However, iso-Aβ_42_ had no effect on the level of the receptor in N2a cells (Figure 2). It is possible that the actions of the intact and modified peptides on the receptor not only differ in the strength of the inhibition but suggests qualitatively different mechanisms of interaction with the receptor. The differences in the action of Aβ_42_ and iso-Aβ_42_ on the α7nAChR level may also be related to the effect on the assembly of receptors inside the cell, as has been shown for other nAChR allosteric modulators of the human Ly6 family [64,65]. 

The molecular modeling data agree with the qualitative differences in the interaction of intact and modified peptides with the receptor. According to the simulation results, the isomerization of the Asp7 residue introduces significant changes into the spatial structure of Aβ_42_ (Figure 4A). These structural changes are reflected in different binding modes of Aβ_42_ and iso-Aβ_42_ to the receptor (Appendix A). In Aβ_42_, the residues surrounding Asp7 and the C-terminal hydrophobic domain are more involved in binding to the receptor, whereas in iso-Aβ_42_ interaction with the α7nAChR is strongly mediated by the residues 11–16, compared to the unmodified Aβ_42_. The model structures of the iso-Aβ_42_ and Aβ_42_ complexes with the α7nAChR are quite similar to each other (Figure 4B), however, the binding of the peptides was different in certain areas. A weaker interaction was predicted for iso-Aβ_42_ with the top-pocket allosteric site (Appendix A) and a stronger interaction with the region of 129–142 of the receptor, the function of which is not known [42]. The prediction of the allosteric nature of the interaction *in silico* is consistent with the data obtained experimentally in vitro, namely with the inhibitory pattern of the amyloid peptides. The differences in the interaction of the peptides with the α7nAChR are also consistent with the differences in the action of the peptides on the intracellular calcium rise mediated by the α7nAChR and with their effects on the expression of the receptor.

It is known that the α7nAChR can mediate the neurotoxicity of Aβ. The α7nAChR modulates fundamental pathways involved in cell survival such as JAK2-STAT3 and the reported binding of β-amyloid to the α7nAChR can effectively decouple the receptors from key pro-survival pathways [14]. Aβ inhibits the response of α7nAChR-containing hippocampal neurons [46], and the neurotoxicity of Aβ decreases with the presence of receptor agonists [38,66,67]. α7nAChR-mediated Ca^2+^ influx directly promotes cell survival [68,69,70], and inhibition of this Ca^2+^ current by Aβ may contribute to neurotoxicity of the peptide.

We decided to test whether the increased ability of iso-Aβ_42_ peptide to inhibit the α7nAChR-mediated Ca^2+^ influx affects its neurotoxic properties. Iso-Aβ_42_ was found to have an increased toxicity for α7nAChR-expressing differentiated SH-SY5Y neuroblastoma cells, which is consistent with the data previously obtained on undifferentiated SH-SY5Y cells or on immortalized H-TERT neurons [47,71]. Importantly, lower concentrations of Aβ neither reproduce toxic effects of such magnitude nor the Aβ-induced loss of nicotinic receptors. These findings suggest that iso-Aβ_42_ toxicity is due to an inhibition of the α7nAChR: the toxic effect of iso-Aβ_42_ is similar in magnitude to that of αBgt, and their co-application does not lead to an increase in cell death (Figure 3). Apparently, both αBgt and iso-Aβ_42_ induce the maximal α7nAChR-mediated level of cell death under these conditions, therefore their effects are not additive. Thus, the increased toxicity of iso-Aβ_42_ may be due to a stronger inhibition of the α7nAChR by this peptide. Some studies report that Aβ toxicity can be mediated by the intracellular Ca^2+^ rise [72,73,74], however, our data supports the protective role of Ca^2+^ entry, at least through the α7nAChR. 

In general, the analysis of the interactions of modified forms of Aβ with the α7nAChR may be crucial for the development of new therapies for AD. In this study, we showed that Asp7 isomerization in Aβ enhances the inhibitory effect of Aβ on the α7nAChR and leads to a more pronounced neurotoxic effect of the peptide. The α7nAChR is widely distributed in the CNS and in the cells of the immune system, while iso-Aβ is one of the most common modifications of Aβ in patients with AD. Thus, to effectively treat AD through the restoration of the α7nAChR function, it may be necessary to target an isomerized form of Aβ, which differs from Aβ in its interaction with the receptor.

## Figures and Tables

**Figure 1 cells-08-00771-f001:**
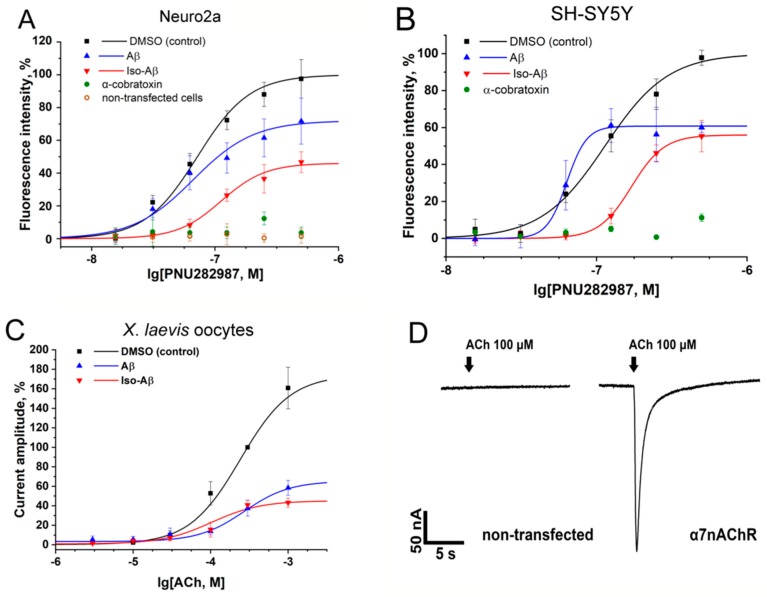
Effects of Aβ_42_ and iso-Aβ_42_ on the intracellular Ca^2+^ and Ba^2+^ rise mediated by the α7nAChR. Dose-dependent Ca^2+^response to PNU282987 stimulation (**A**) in α7nAChR-transfected or in non-transfected N2a and (**B**) in differentiated SH-SY5Y cells pre-incubated with 10 µM of Aβ_42_ or iso-Aβ_42_ for 30 min or with 15 µM of α-cobratoxin for 15 min. (**C**) Dose-dependent Ba^2+^ response in the α7nAChR-transfected *Xenopus laevis* oocytes to acetylcholine (ACh) stimulation after 3 min pre-incubation with 10 µM of Aβ_42_ or iso-Aβ_42_. (**D**) Electrophysiological recordings of α7nAChR currents in native (uninjected) and α7nAChR cRNA-injected oocytes. Currents were obtained in response to 100 μM ACh application (arrows). Vertical bar represents current scale (50 nA). Data are presented as the mean ± SEM of at least three independent experiments.

**Figure 2 cells-08-00771-f002:**
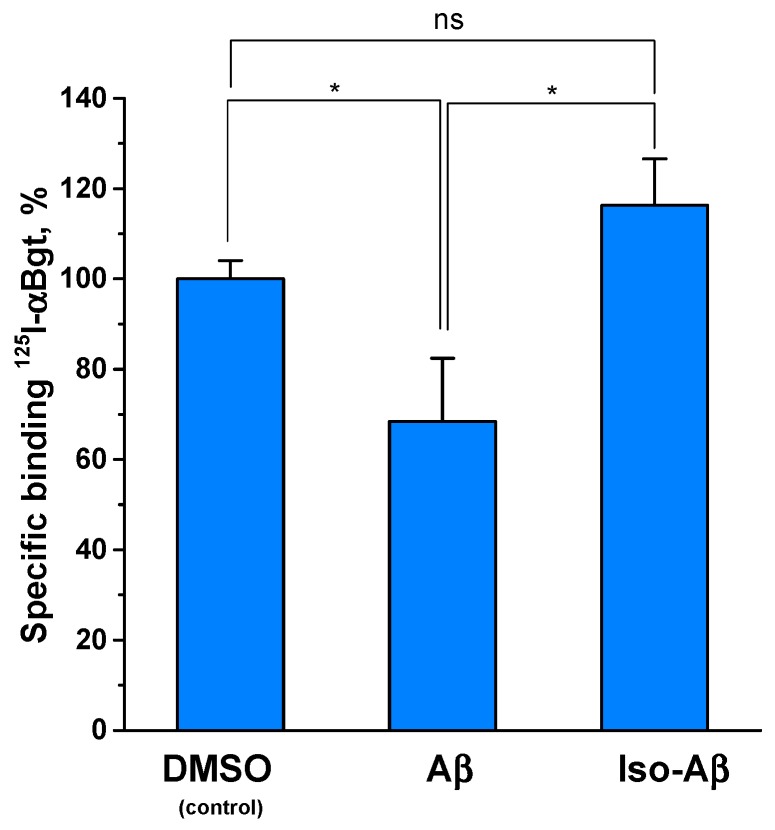
Specific binding of ^125^I-αBgt to Neuro2a cells transiently transfected with human α7nAChR. The cells were pre-incubated for 72 h with either 0.4% DMSO (control) or 10 μM of Aβ_42_ or iso-Aβ_42_ dissolved in DMSO. Specific binding in the control is set to 100%. Data are presented as the mean ± SEM of two independent experiments with at least five replicates for each point. * *p* < 0.05, ns = non-significant.

**Figure 3 cells-08-00771-f003:**
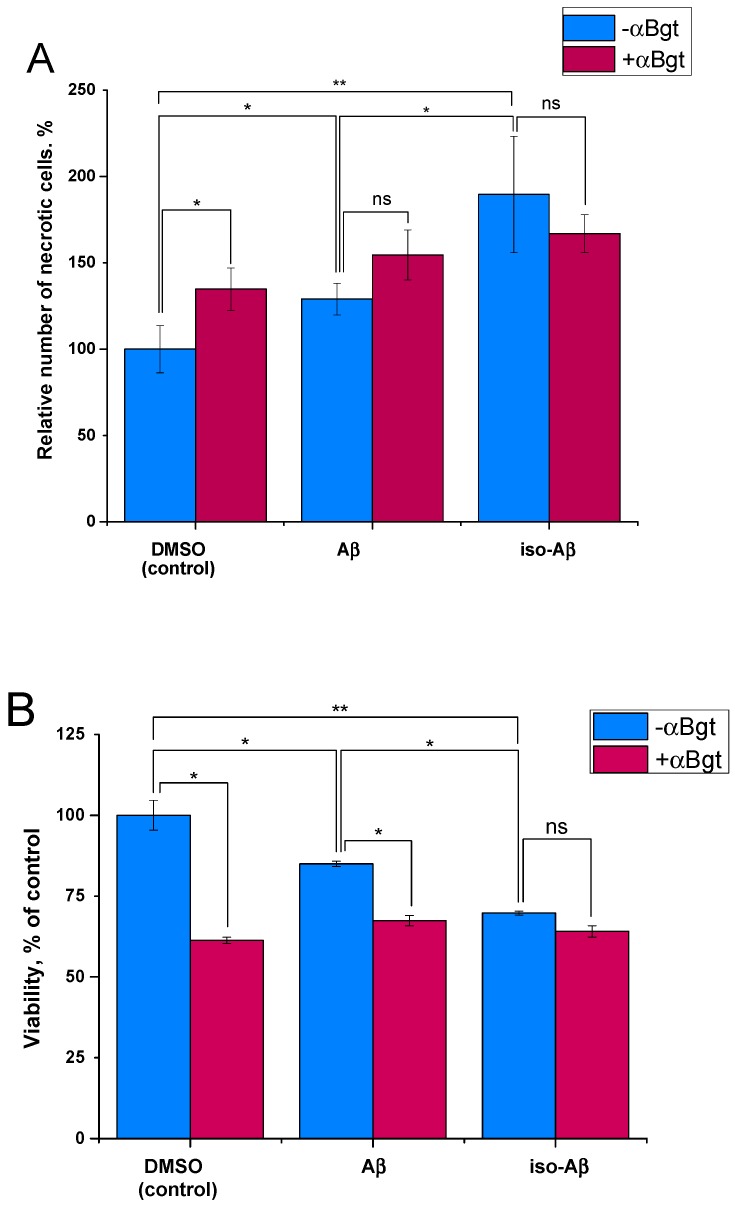
Neurotoxic effects of Aβ_42_ and iso-Aβ_42_ on differentiated SH-SY5Y cells were measured after 72 h of incubation with 10 µM of peptides in the presence or absence of α-bungarotoxin (αBgt) (50 nM). (**A**) Relative number of necrotic cells—represented by normalized Ethidium dimer (EthD-1) fluorescence. (**B**) Viability of cells in relation to the control—measured with the WST test. Data are presented as the mean ± SD of two independent experiments with at least five replicates for each point. * *p* < 0.05, ** *p* < 0.01, ns = non-significant.

**Figure 4 cells-08-00771-f004:**
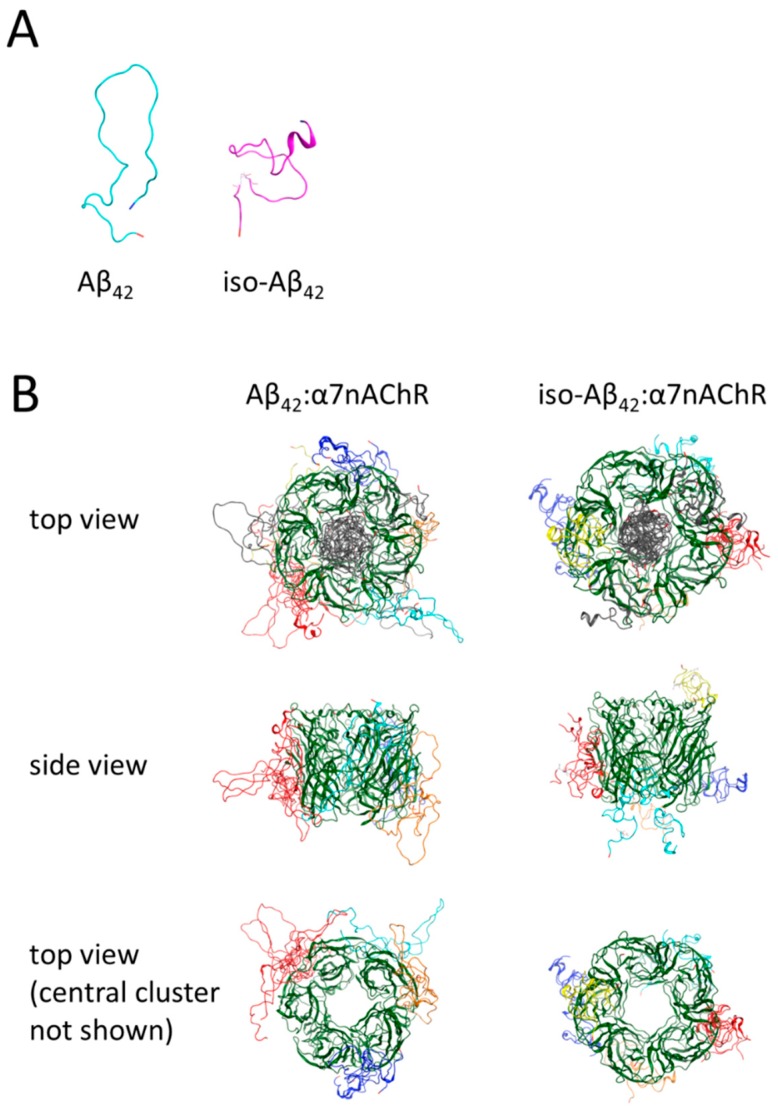
(**A**) Models of Aβ_42_ and iso-Aβ_42_ in solution. (**B**) Models of the α7nAChR complexes with the amyloid peptides.

**Table 1 cells-08-00771-t001:** The half maximal effective concentration (EC50) of PNU282987inducing the α7nAChR–mediated intracellular calcium rise in N2a and SH-SY5Y neuroblastoma cell lines, treated with Aβ_42_ oriso-Aβ_42_. Data are presented as the mean of at least three independent experiments ± SEM.

Cell Line	Treatment	EC50 Value, nM
N2a	Control	71.9 ± 8.4
	Aβ_42_	66.8 ± 8.6
	iso-Aβ_42_	112.6 ± 4.3
SH-SY5Y	Control	114 ± 7.0
	Aβ_42_	63.4 ± 2.3
	iso-Aβ_42_	171 ± 3.3

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
