# Peer review of "Isomerization of Asp7 in Beta-Amyloid Enhances Inhibition of the α7 Nicotinic Receptor and Promotes Neurotoxicity"

_cells, 2019, doi:10.3390/cells8080771_

Round 1
Reviewer 1 Report
This manuscript describes the effect of Ab42 and iso-Ab42 in different model cell lines in terms of alteration of Ca2+ responses that are mediated by the a7nAchR, either endogenously expressed or upon over-expression of the human- a7nAchR. In addition, it investigates, by computer modeling, the interaction of the two Ab peptides with this receptor.
The Authors conclude that the two peptides exert different effect on Ca2+ signaling mediated by a7nAchRs, as well as on cell toxicity and viability, being the iso-Ab42 more effective in reducing Ca2+ currents and causing cell death. By this conclusion the Authors suggest that AD therapy should also target isomerization forms of Ab for being successful.
Altogether the presented data raise the interesting issue of the different roles exerted by Ab42 and iso-Ab42 on cell viability and their effect of Ca2+ signaling. Whereas the data obtained on Ca2+ responses and cell death are clear, the linkage with the experiments in oocytes and the analysis by computer modeling is rather weak. The following points need to be better investigated or properly explained.
1. Ab peptides have high affinity for the a7nAchR, but the Authors use very high concentrations of both the peptides, likely causing effects also on other receptors. Which is the rationale of using 10-15 uM concentrations? What is the lowest concentration that allows to reproduce one or more of the described effects?
2. Which is the ratio of the two peptides found in AD samples? How much do Ca2+ responses and toxicity depend on this ratio? Please provide more details on physio-pathological levels of iso-Ab42. How do Ca2+ responses depend on this ratio?
3. The Authors use very different protocols for Ab exposure in the different assays. They should try to interpret the results accordingly: for instance Ca2+ responses are evaluated after a 30 min incubation whereas oocytes’ currents only after 3 min. For receptor internalization and cell death a 72 h incubation is employed, however the first assay is carried out in culture medium while the second one in a serum-free medium. How do the presence of serum change the effective Ab concentration?
4. The data obtained with oocytes are markedly different from those obtained in N2A or SH-S5-5Y cells: no clear difference was found between Ab42 and iso-Ab42 in oocytes with only 3-minute-incubation. This finding is substantially in agreement with the data obtained by computer modeling since Authors state. “…the model structures of the iso-Aβ42 and Aβ42 complexes with the α7nAChR are quite similar to each other …”.One possible interpretation of these results is the fact that in oocytes the Authors substantially measure the pure effect of the two peptides on the a7nAchR conductance without secondary effects due to receptor internalization, alteration of membrane potential, or induced by Ca2+ entry. Note that the oocyte experiments are carried out in voltage-clamp and use Ba2+ instead of Ca2+ . For this reason no acute effect due to Ca2+ entry, such as, fast/slow current inactivation and/or receptor desensitization will occur. Moreover also oligomerization might be different in a Ba2+ containing medium. The Authors should make this point clear in the text as well as in the figure legends: what they measure are Ca2+ rises in N2A and SH-SY5Y cells, that are a balance of Ca2+ entry, release and extrusion (not simply Ca2+ currents) whereas in oocytes they measure Ba2+ currents that are a surrogate of Ca2+ currents.
5. How much the differences between the Ca2+ responses measured in N2A and SH-SY5Y cells depend on the different Ca2+ probes used (in terms of dynamic ranges and Kds)? Please show average or typical Ca2+ traces
6. How much do the Ca2+ responses depend on the presence of other subunits in the receptor composition, especially in SH-SY5Y where it is expressed endogenously?
Minor points:
Some methodological aspects are missing or not clearly explained:
7. The authors state that …The higher neurotoxicity of iso-Aβ42 compared to that of Aβ42 was not due to its enhanced aggregation or differences inoligomeric composition, as we demonstrated by the electrophoretic analysis of covalently linked oligomers obtained by photo-induced crosslinking (Fig. S2).There is no explanation of the methodology used for measuring monomers and oligomers. Please provide methods and a figure of the electrophoretic analysis.
8. Please give the details of SH-SY5Y differentiation.
9. Please give the details of GH4C1 cell culture.
10. Why do the Authors use paired t-testand not unpaired?
11. Fig. 3 and Fig. 4 have the same legend on y-axis but they measure different things (necrotic cells and cell survival).
12. Figures with not statistical significant changes (see Fig. 2) might be summarized in the text or in a table, if required.
13. Panels within figures should have a title indicating the cell line
Reviewer 2 Report
The authors demonstrated that the isomerization of Asp7 enhances the inhibitory effect of Aβ on the functional activity of the α7nAChR, which may be an important factor in the disruption of the cholinergic system in AD. This manuscript shows some interesting findings. However, several questions are needed to be clarified before it can be considered for publication.
Major comments
(i) The negative controls are needed for Figure 1 to specify these signals are α7nAChR-dependent calcium responses. The data of α7nAChR RNAi or inhibitor in Figure 1A, and the data of non-transfected cells in Figure 1B and C should be added.
(ii) The authors described that the data in Figure 3 indicating a decrease in the α7nAChR levels in the plasma membrane of N2a cells. To prove a decrease in the α7nAChR levels, the Westernblotting of α7nAChR in plasma membrane fraction should be performed.
(iii) The authors suggest the binding modes between Aβ42/iso-Aβ42 and the receptor is different that can possibly explain the previous experiments. If possible, please explain the reason of different binding modes from view of structural and residual differences of Aβ42/iso-Aβ42 peptides, as there may be common (chemical) mechanism why iso-Aβ42 could or could not bind to (specific) amino acids, in contrast to Aβ42. Please refocus on the residues of Aβ42 peptides that are interacting with the receptor, to precisely understand the underlying mechanism.
Minor comments
(i) The vertical axis of Figure 4A would be "Relative number of necrotic cells", not "Viability".
(ii) The sentence of line 198-200 should be deleted.
Round 2
Reviewer 2 Report
Major points
(i) The data of non-transfected cells in Figure 1A (N2a) and C (Xenopus laevis oocytes) should be added. It is needed to show whether the background levels of calcium responses will be observed in non-transfected cells with or without α-cobratoxin.
(ii) My main concern is that there is no evidence that α7nAChR protein was over-expressed in Neuro2a and Xenopus laevis oocytes in this paper. The Western blotting of α7nAChR should be performed in non-transfected and α7nAChR-transfected cells. You can detect the increased band in α7nAChR-transcfected cells compared to that of non-transfected cells if the transfection is succeeded.
